# Influence of the Synthesis Method on the Structural Characteristics of Novel Hybrid Adsorbents Based on Bentonite

**Dariusz Sternik [1], Mariia Galaburda [2],\* , Viktor M. Bogatyrov [2] and Volodymyr M. Gun'ko [2]**

[1]   Faculty of Chemistry, Maria Curie- Sklodowska University, Maria Curie-Sklodowska Sq. 3, 20-031 Lublin, Poland; dsternik@poczta.umcs.lublin.pl

[2]   Chuiko Institute of Surface Chemistry, NAS of Ukraine, 17 General Naumov Str., Kyiv 03164, Ukraine; vbogat@ukr.net (V.M.B.); vlad_gunko@ukr.net (V.M.G.)

**\***   Correspondence: mariia.galaburda@gmail.com; Tel.: +38-044-422-9672

**Abstract:** New hybrid composite materials were prepared by polymerization of resorcinol–formaldehyde resins in the presence of bentonite with various contents of polymer and water, and then exposed to pyrolysis in an inert atmosphere at 800 $^{\circ}$C. The influence of the filler and synthesis method on the morphological, textural and structural characteristics has been described. The materials were characterized using low temperature nitrogen adsorption–desorption, small angle X-ray scattering, scanning electron microscopy, Raman spectroscopy, differential scanning calorimetry (DSC), and thermogravimetry analysis (TGA). The maximal values of the specific surface area of organo-bentonite and carbonized samples were 254 and 200 $m^2$/g, respectively, which is much larger than that of the initial bentonite. The TGA and DSC experiments showed changes in the thermal stability of samples depending on their composition. The obtained data could provide a better understanding of the principles of preparing hybrid bentonite-containing composites that may provide an additional incentive to develop advanced technologies.

**Keywords:** resorcinol–formaldehyde; pyrolysis; bentonite/carbon composites; thermal degradation

## 1. Introduction

Bentonite, a layered clay mineral mainly composed of montmorillonite, is one of the promising sorbents to remove dye pollutants from wastewaters because of appropriate textural properties [1], abundance [2], high chemical and mechanical stability, and low cost [3]. The application areas of bentonites vary depending on the amounts of their constituents. They can be used as selective adsorbents [4] of dyes [5] and herbicides [6], as well as catalysts [7]. The use of bentonite as reinforcing and viscosity modifiers in polymer composites and drilling fluids was described in detail [8,9]. Its high capacity to remove excess water from some powders allows the use of bentonite as a binding agent in pelletization of iron ore [10]. Furthermore, natural clays are readily available materials performing as excellent cation exchangers [11]. Bentonite is easily dispersed in water and has good adsorptive properties [12]. Clays can remove a wide range of water contaminants because of micro-sized particles with relatively large external and internal surface areas [13]. However, the difficulties encountered in extracting clay particles from solutions after the adsorption process make them less attractive as sorbents for industrial water purification. This problem ultimately makes the regeneration of these colloidal particles quite difficult. Additionally, clay minerals have a poor affinity to low molecular weight organics because the hydration of clays results in diminution of the accessibility of interlayer spaces, e.g., for aromatic molecules. The clay minerals lose a significant part of the adsorption capacity after regeneration. Recently, several approaches for bentonite modification were studied, e.g.,

incorporation of chitosan [14], calcium alginate [15], activated carbon [16], hydrochar [17], polymerized 4-vinylpyridine [18], magnesite [19], and polyaniline [20] into the bentonite networks to enhance the sorption properties.

Bentonite is used for the synthesis of novel superabsorbent polymer composites [21,22]. Preparation of clay-based absorbents as polymer composites is of importance due to improved properties and lower cost of the resulting materials compared to that of pure polymers. Recently, the use of organo-bentonites was investigated to improve organic contaminant sorption [23,24].

There is a growing interest in resorcinol–formaldehyde (RF) resins which are among the most famous organic gels [25]. These resins are easily synthesized by condensation of resorcinol and formaldehyde in an aqueous solution (e.g., at a molar ratio of 1:2). The structural/adsorption characteristics of the RF polymers and polymer-based composites can be controlled by changing synthesis conditions such as resorcinol-to-formaldehyde, resorcinol-to-water, resorcinol-to-catalyst ratios, and pH value. During the synthesis of RF-bentonite composites, small molecules of the initial monomers dissolved in water can penetrate into pores of bentonite that affect the structural characteristics of organic-clay and related char-clay composites. In addition, surface compounds of alkali and alkaline-earth metals in bentonite have a catalytic effect on polymer synthesis and subsequent carbonization. Therefore, changes in the ratio of components and synthesis conditions allow us to control the structural characteristics.

The main objective of this work was to investigate the influence of the synthesis method on the changes in the structure, surface area and porosity of resorcinol–formaldehyde/bentonite and related char/bentonite composites. Pyrolysis of polymers bound to inorganic particles allows us to fabricate nanostructured hybrid materials retaining the structural advantages of bentonite (mesoporosity) and adding new structural characteristics due to a carbon component.

## 2. Materials and Methods

### 2.1. Materials

The experiments were conducted using bentonite (Sigma–Aldrich, Cat.: 28,523-4, Lot.: STBB6144, CAS:1302-78-9, Gillingham, UK); resorcinol (99.9%, Chimlaborreativ, Ukraine), and a 37% aqueous solution of formaldehyde (GOST 1625-89, Chimlaborreativ, Ukraine).

Two series of organo-clays were synthesized. The first one consisted of three samples prepared by mixing (with a magnetic stirrer) bentonite (10 g), resorcinol (10, 6 or 3 g), formalin (15, 9, or 4.5 g), and distilled water (0, 6, or 11.3 g) until the complete dissolution of resorcinol. The reaction took place in a plastic container with a lid in the desiccator for 2.5 h. As a result, hard light-brown gel was formed. Solid cylinders of the polymer composites were mechanically crushed and dried at 110 °C for 2 h. The samples were labelled as RFR-75, RFR-76, and RFR-77, respectively.

At the next stage, all samples were pyrolyzed (at 800 °C for 2 h) in quartz cells of a reactor (at a heating rate of 10 °C/min) with an argon flow (100 mL/min). Cooling took place in a stream of argon at room temperature. The samples were labelled as RFC-75, RFC76, and RFC-77, respectively.

The second series of samples was prepared by mixing formalin (15 g) and bentonite (10 g) in plastic containers. The suspension was kept for swelling in formalin for 5 days at room temperature, with periodic stirring. Then resorcinol (10, 6, or 3 g) was added under stirring. The mixture was left at room temperature for gel formation. After seven days, all samples were hardened and had a brown color. Solid cylinders were crushed and dried at 120 °C for 5 h. The samples were marked as RFR-78, RFR-79, and RFR-80, respectively. The samples in each series differed in the mass ratio of resorcinol to formalin. The pyrolysis was carried out similarly as in the first series. The samples were labelled as RFC-78, RFC-79, and RFC-80, respectively.

A control RFR sample was prepared by stirring of resorcinol (10 g) with an added solution of formaldehyde (15 g dissolved in 10 g of water). This mixture was hermetically sealed, placed in a thermostatic oven, and treated at 85 °C for 2 h. After gelling, a brown solid polymer composite was

obtained, which was dried at 85 °C for 18 h. Then it was milled and sieved in order to obtain a 0.50 mm fraction. The pyrolysis was carried out similarly to the RFR/bentonite series.

A detailed ratio of the initial components used to obtain composites can be seen in the electronic supplementary material (ESM) file (Table S1).

*2.2. Methods*

A scanning electron microscopy (SEM) study was performed using a Quanta$^{TM}$ 3D FEG (FEI, Hillsboro, OR, USA) operating at a voltage of 30.0 kV.

To analyze the textural characteristics, low-temperature (77.4 K) nitrogen adsorption–desorption isotherms were recorded using a Micromeritics ASAP 2405N adsorption analyzer. The specific surface area ($S_{BET}$) was calculated according to the standard BET method (using Micromeritics software). The total pore volume, $V_p$, was evaluated from the nitrogen adsorption at $p/p_0$ = 0.98–0.99 ($p$ and $p_0$ denote the equilibrium and saturation pressure of nitrogen at 77.4 K, respectively). The nitrogen desorption data were used to compute the pore size distributions (PSD, differential $f_V(R) \sim dVp/dR$ and $f_S(R) \sim dS/dR$), using a model with slit-shaped and cylindrical pores and voids between spherical nanoparticles (SCV) with a self-consistent regularization (SCR) procedure for organo-bentonite samples and slit-shaped pores for carbon-bentonites [26,27]. The differential PSD with respect to pore volume, $f_V(R) \sim dVp/dR$ and $\int f_V(R)dR \sim V_p$, were recalculated as incremental PSD (IPSD), $\sum \Phi_{v,i}(R) = V_p$. The $f_V(R)$ and $f_S(R)$ functions were also used to calculate the contributions of micropores ($V_{micro}$ and $S_{micro}$ at radius $R \leq 1$ nm), mesopores ($V_{meso}$ and $S_{meso}$ at 1 nm < $R$ < 25 nm) and macropores ($V_{macro}$ and $S_{macro}$ at 25 nm < $R$ < 100 nm) to the total pore volume and specific surface area. Additionally, for some samples, nonlocal density functional theory (NLDFT) and (DFT) [27] methods were used.

The samples were analyzed using the small angle X-ray scattering (SAXS) method with an Empyrean diffractometer (PANalytical), CuK$_\alpha$ radiation of $\lambda$ = 0.15418 nm over the 0.5–5° range of 2θ (details of SAXS data treatment are shown in the ESM file).

The Raman spectra were recorded over the 3200–150 cm$^{-1}$ range using an inVia Reflex Microscope DMLM Leica Research Grade, Reflex (Renishaw, UK) with laser excitation at $\lambda_0$ = 785 nm.

The thermal stability of samples was determined using a STA 449 Jupiter F1 (Netzsch, Germany) coupled online with a QMS 403D Aëolos (Netzsch, Germany) mass spectrometer and a FTIR spectrometer (Bruker, Germany). The samples (~12 mg) were heated at a rate of 10 °C/min in the range of 30–950 °C in the atmosphere of synthetic air (flow of 50 mL/min) in an alumina crucible, and sensor thermocouple type S TG–DSC. An empty Al$_2$O$_3$ crucible was used as a reference. Thermogravimetry (TG and DTG curves), differential scanning calorimetry (DSC) and FTIR spectra of the gaseous products were registered during analysis. Data were collected and processed using the NETZSCH Proteus$^{®}$ software (version 6.1).

## 3. Results and Discussions

The structural characteristics of nanocomposites were studied using low-temperature nitrogen adsorption–desorption isotherms (see Figure S1 in ESM file). All of these materials exhibited the nitrogen adsorption isotherms of type IV (H3 type of hysteresis loops) according to the IUPAC classification [28]. Capillary condensation occurred in a relatively wide range of pressures at $p/p_0$ = 0.45–1.0, which showed the presence of mesoporosity at broad PSD. Typically, an increase in a general slope of the curves (Figure S1) corresponds to an increase in contribution of larger mesopores.

The BET surface area (Table 1, $S_{BET}$), as well the $S_{SAXS}$ values (Table S2), of untreated polymer-bentonite nanocomposites of the first series were significantly higher than that of the second one. This difference was especially visible for RFR-75 and RFR-78, which have the same ratio of components but differ in the method of synthesis (ESM file, Table S1). This is due to the increased density and/or blocking of mesopores. For example, the $S_{SAXS}$ values (total surface area of open and closed pores) are typically larger than the $S_{BET}$ values (open area accessible for $N_2$ molecules). The BET surface area of the nanocomposites of the first series (RFx75-77) decreased after pyrolysis

from ≈240 m²/g to ≈170 m²/g, but for the second series, it increased from ≈80 m²/g to ≈210 m²/g. The pore size distributions (Figure 1 and Table 1) are of a complex shape due to the presence of different components in the composites that are characterized by different textural and morphological features.

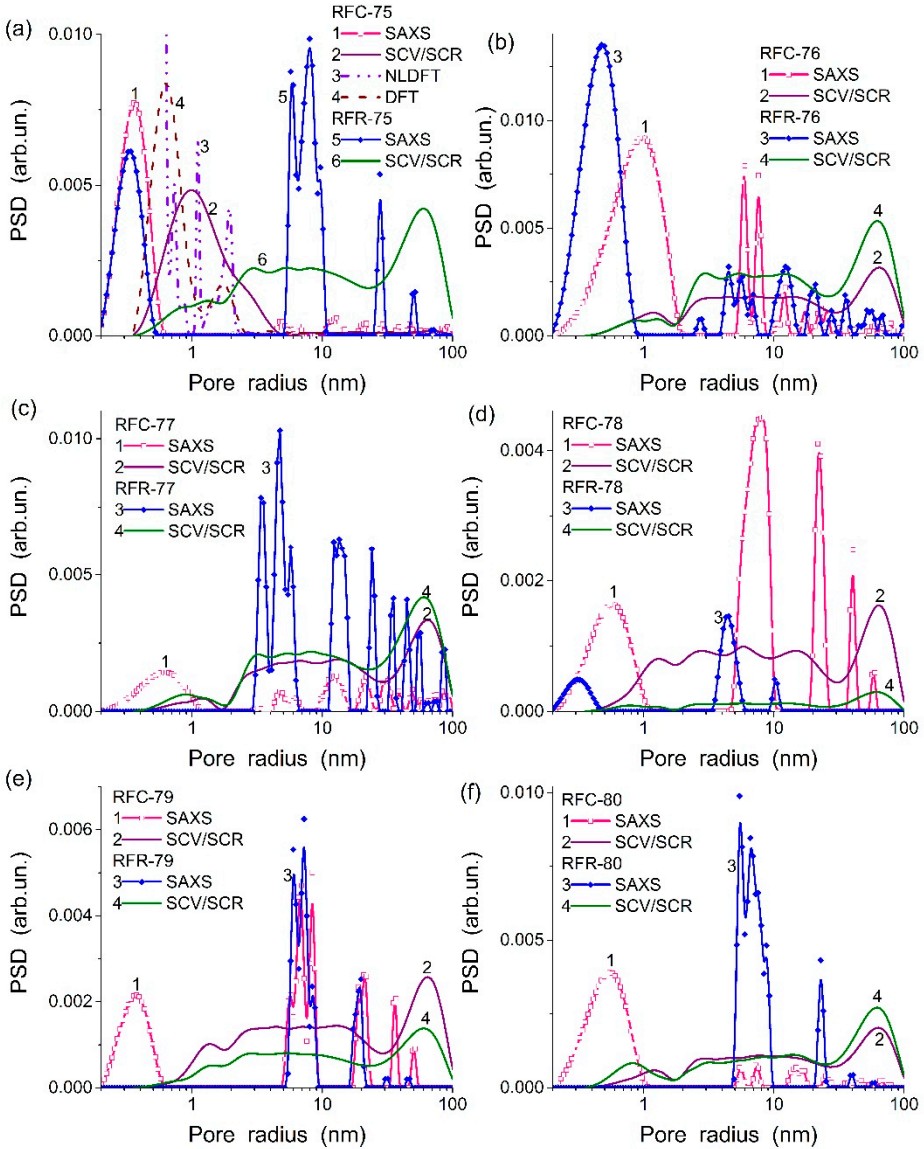

**Figure 1.** Pore size distributions calculated using nitrogen adsorption–desorption isotherms (using (SCV/SCR), nonlocal density functional theory (NLDFT) and (DFT) methods) and small angle X-ray scattering (SAXS) data for (**a**) RFR-75 and RFC75, (**b**) RFR-76 and RFC76, (**c**) RFR-77 and RFC-77, (**d**) RFR-78 and RFC-78, (**e**) RFR79 and RFC-79, and (**f**) RFR-80 and RFC-80.

Note that the SAXS PSD differ from the SCV/SCR PSD (Figure 1) due to (i) the contribution of closed pores in the first ones, (ii) the difference in the regularization parameters (it was smaller upon the SAXS PSD calculations), and (iii) certain differences in sample pre-treatment before the measurements and upon mathematical treatments of the data.

The obtained results show that the textural characteristics depended strongly on the synthesis methods. The organo-bentonite composites of the first and second series are significantly different due to features of penetration of organic molecules of formaldehyde and resorcinol into the pores and interlayer space of bentonite. For the first series, the synthesis of organo-bentonite was carried out rather quickly in comparison with the second series and a polymer phase could totally cover bentonite

particles and could only partially cover the inner surface of pores available for rapid penetration of the aqueous solution of organic components. Hereby, for this series, resorcinol–formaldehyde polymer was formed on bentonite. The volume of micropores and mesopores has considerably increased in comparison with bentonite (RFR-75-77, Table 1). The high-temperature carbonization of RFR leads to changes in the structure of bentonite and the reorganization of the carbon skeleton. Consequently, these processes led to changes in the structural characteristics of the carbon-inorganic composite.

The synthesis of the second series was characterized by long contact time of bentonite with aqueous solutions of formaldehyde and resorcinol that can provide more effective penetration of components into the interlayer space of bentonite. As a result, the specific surface area and total pore volume of the organo-bentonites (RFR-78-80) were smaller than that of pure bentonite (Table 1). In the pores of bentonite, a weakly porous polymer could be formed, since, according to the synthesis conditions, the molar ratio of resorcinol–formaldehyde was changed over a wide range. This did not contribute to the formation of pores in the polymer. During subsequent high-temperature pyrolysis of RFR, micropores were formed that were accompanied by an increase in the specific surface area.

**Table 1.** Structural characteristics of tested samples.

| Sample | $S_{BET}$ $(m^2/g)$ | $V_p$ $(cm^3/g)$ | $V_{micro}$ $(cm^3/g)$ | $V_{meso}$ $(cm^3/g)$ | $V_{macro}$ $(cm^3/g)$ | $V_{micro}/V_p$ | $V_{meso}/V_p$ |
|---|---|---|---|---|---|---|---|
| Bentonite | 86 | 0.117 | 0.013 | 0.087 | 0.017 | 11.4 | 74.1 |
| RFR-75 | 253 | 0.183 | 0.060 | 0.118 | 0.005 | 32.5 | 64.6 |
| RFR-76 | 245 | 0.215 | 0.043 | 0.167 | 0.005 | 20.0 | 77.8 |
| RFR-77 | 160 | 0.163 | 0.028 | 0.125 | 0.010 | 17.1 | 77.0 |
| RFC-75 | 199 | 0.113 | 0.080 | 0.031 | 0.002 | 71.1 | 27.3 |
| RFC-76 | 175 | 0.112 | 0.045 | 0.065 | 0.002 | 39.9 | 57.9 |
| RFC-77 | 122 | 0.129 | 0.033 | 0.091 | 0.005 | 25.6 | 70.8 |
| RFR-78 | 14 | 0.016 | 0.011 | 0.004 | 0.001 | 70.7 | 24.0 |
| RFR-79 | 84 | 0.062 | 0.029 | 0.031 | 0.002 | 47.2 | 50.1 |
| RFR-80 | 82 | 0.095 | 0.025 | 0.065 | 0.005 | 26.5 | 67.9 |
| RFC-78 | 208 | 0.11 | 0.064 | 0.043 | 0.003 | 58.2 | 39.5 |
| RFC-79 | 134 | 0.085 | 0.054 | 0.028 | 0.003 | 64.0 | 32.8 |
| RFC-80 | 95 | 0.083 | 0.032 | 0.048 | 0.003 | 38.1 | 58.0 |

In order to elucidate the morphology features of carbonized composites, scanning electron microscopy was used (Figure 2 and Figure S4). Non-carbonized samples of the first series demonstrate disperse structures with agglomerates of irregular shapes, whereas the samples of the second series are denser and formed flat, extended agglomerates. After the carbonization process, visual changes in the surface morphology are rather insignificant (Figure 2 and Figure S4). However, for the first series, the samples exhibited a slightly rough surface, whereas for the second series, the formation of more layered structures took place. These morphological and textural features of the samples were well seen in the distribution functions of chord sizes (i.e., pore wall thickness) (Figure S3a) and particle size distributions (Figure S3b) with contributions of lamellar, cylindrical, and spherical particles (Table S2, $c_{lam}$, $c_{cyl}$, and $c_{sph}$ values, respectively, calculated using the SCR method). However, there were certain differences in the sample morphology at a microlevel (SEM) and a nanolevel (SAXS, nitrogen adsorption).

Additional structural information on the carbon phase in composites can be obtained using Raman spectroscopy. The spectra were collected in the most informative range for carbon materials of 200–3200 cm$^{-1}$ (Figure 3). Two typical first order bands appeared, i.e., the G (located at ~1600 cm$^{-1}$) and the D (at ~1320 cm$^{-1}$) bands. The ratio between the integral intensities of the G and D bands ($A_G/A_D$ ratio as a measure of the graphitization degree, i.e., contribution of plane polyaromatic structures) is an indicator of the crystallinity degree [29]. The values of $A_G/A_D$ and FWHM of the G band were calculated by deconvolution of the spectra using Lorentzian functions. A relatively low G/D ratio

(~0.9) indicated the formation of the carbon phases with disordered structures, and it showed that graphitic layers are semi-crystalline and possess many defects that are typical for chars prepared at relatively low temperatures (800 °C). Additionally, the formation of a sharp G band (*FWHM_G* is ~54 cm$^{-1}$) specifies the development of carbon particles (graphenes) of small sizes.

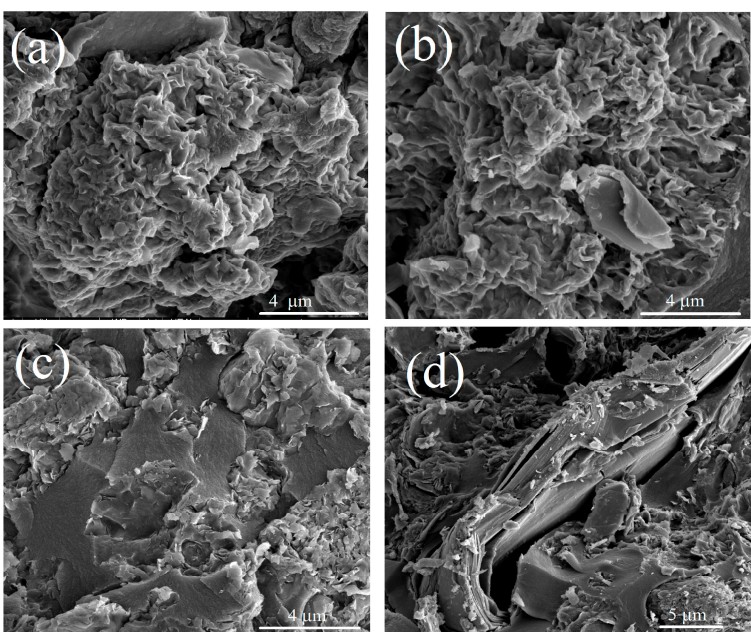

**Figure 2.** Sample morphology at microlevel (SEM) images of untreated nanocomposites RFR-76 (**a**), RFR-79 (**c**), and carbonized RFC-76 (**b**), RFC-79 (**d**) (scale bar 4–5 μm).

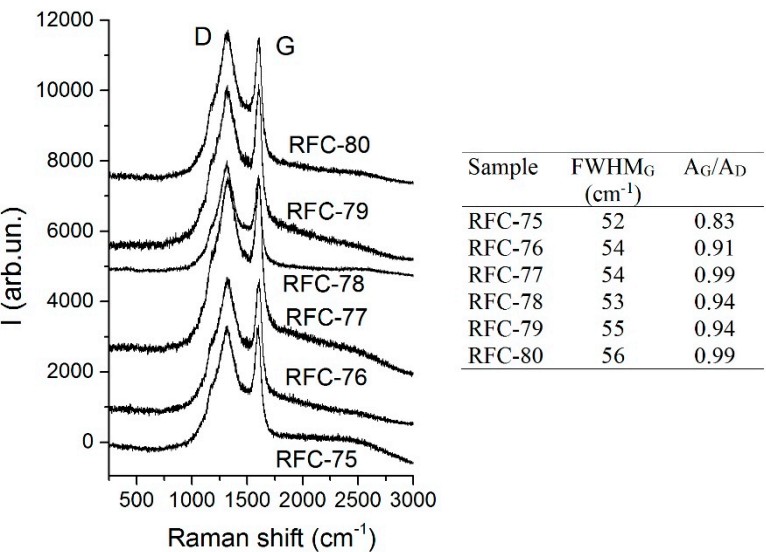

| Sample | FWHM$_G$ (cm$^{-1}$) | A$_G$/A$_D$ |
|---|---|---|
| RFC-75 | 52 | 0.83 |
| RFC-76 | 54 | 0.91 |
| RFC-77 | 54 | 0.99 |
| RFC-78 | 53 | 0.94 |
| RFC-79 | 55 | 0.94 |
| RFC-80 | 56 | 0.99 |

**Figure 3.** Raman spectra of the nanocomposites.

The thermogravimetric analysis (TG/DTG) combined with differential scanning calorimetry (DSC) and recording of the FTIR spectra of the gaseous products was used to analyze thermal changes which occurred in various samples depending on their composition. The TG/DTG and DSC curves of the initial bentonite and organo-bentonite samples strongly differ (Figure 4). For bentonite, three endothermic peaks were observed: a dehydration stage followed by two dehydroxylation stages. The first dominant mass loss (4.64% at maximum at 125.2 °C) was due to the removal of the adsorbed and interlayer water. The next stages (mass loss 6.19%) were due to dehydroxylation in the temperature

ranges of 400–550 °C and 550–750 °C, and the corresponding endothermic peak with a maximum at 489 °C and 681.8 °C. There are several causes of this two-step dehydroxylation [30]. For composites, the final residue is consistent with a mixture of ash and carbon. Concerning the control polymer sample, oxidative degradation of crosslinked resorcinol–formaldehyde proceeded in two main stages (Figure 4b). In the first stage, up to 200 °C, water was removed (mass loss of 11%). In the second stage, 88 wt.% of polymer was removed at 200–800 °C, with maximum removal at ~540 °C.

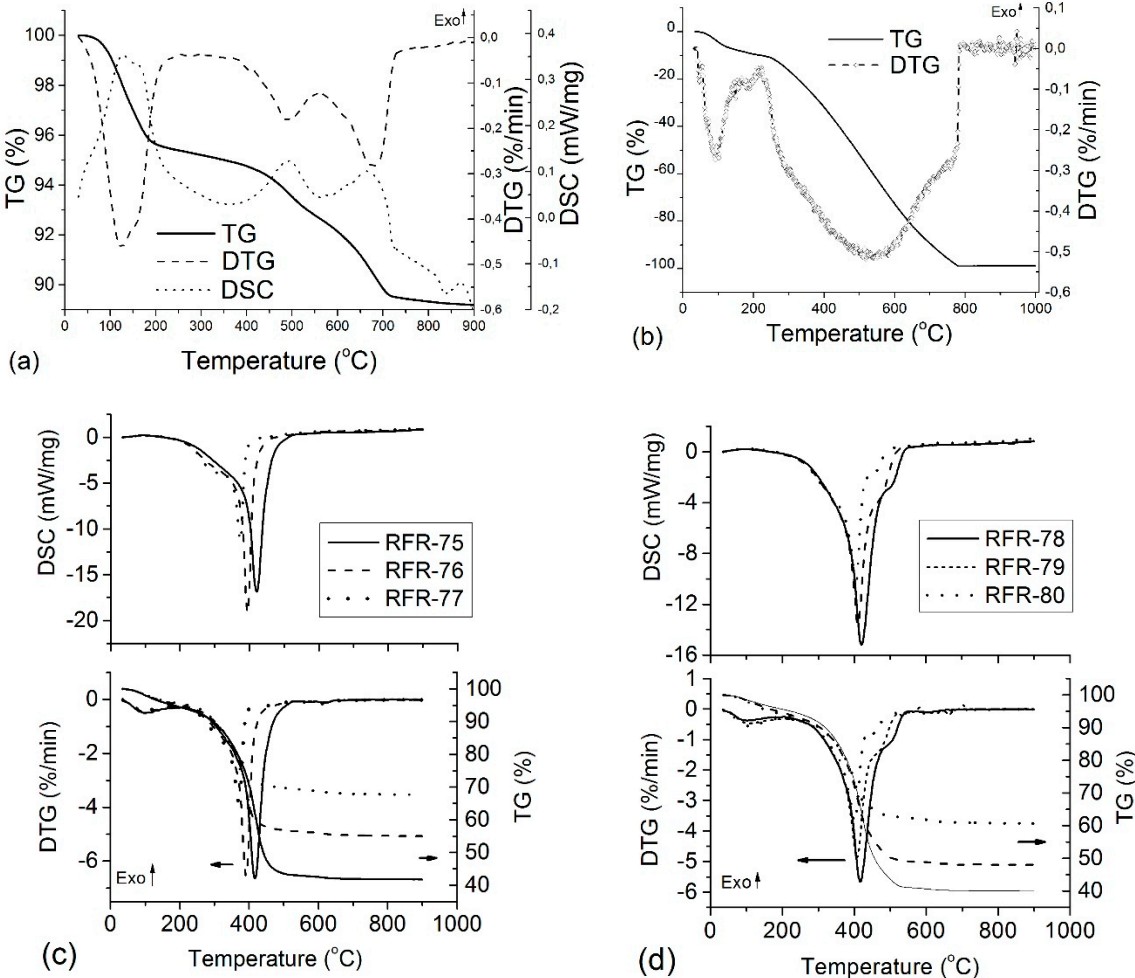

**Figure 4.** TG, DTG, and differential scanning calorimetry (DSC) curves of initial bentonite (**a**), control polymer (**b**) and organo-bentonite samples of first (**c**) and second (**d**) series.

TG, DSC, and DTG curves of composites of both series (Figure 4c,d) show two main weight losses assigned to dehydration (50–200 °C) and complex dihydroxylation of bentonite with simultaneous oxidative decomposition of the polymer with two overlapping stages at 200–600 °C. These weight losses, caused by scission of C–C linkages, were maximal at 360–416 °C. At $T > 250$ °C, carbon oxidation and removal of volatile $CO_2$ occurred with the formation of carbonaceous residue, which decomposed slowly up to 800 °C. For composites, the temperature range of polymer degradation become narrower in comparison to control polymer. This behavior can be explained by several reasons. First, catalytic degradation of polymer occurred in the presence of bentonite, causing a 100 °C decrease in decomposition temperature compared to the initial polymer. For example, Florencio et al. reported that the thermal degradation of polypropylene (in the solid state) filled by a clay was stronger than that of the pure polymer [31]. Secondly, the rate of thermal-oxidative degradation of organic material depends on the possibility of oxygen penetration to the surfaces of polymeric hydrocarbon structures.

Clearly, in the hybrid organo-bentonite composites, polymer structures are less accessible for oxygen molecules. The specific surface of organo-bentonite samples was significantly higher than that of the initial polymer (250 m$^2$/g vs. 25 m$^2$/g). In this situation, the complete oxidation of the organic polymer will occur faster that can be seen from the TG, DTG, and DSC data.

Gram–Schmidt curves (Figure S5) confirm the maximum release of gaseous products of the thermal decomposition of composites in the range of 200–600 °C. The position and change in their intensity coincide with the DTG curves. The amount of physically-adsorbed water was less in the composites than that in bentonite and polymer alone. A slight shift of the peaks related to region II toward the region of high temperatures for the second series of composites in comparison to the first series was due to the formation of denser organo-bentonite agglomerates. A relatively high quantity of ash was attributed to the presence of some inorganic compounds. An increase in the resorcinol content during the synthesis enlarged the polymer content in the final composites. However, there was a difference in the polymer content in composites with an equal ratio of resorcinol/bentonite at the output of the carbonized product of the first and second series (Table 1). For samples of the first and second series, the molar ratio of resorcinol/formaldehyde was 1/2 and 1/6.6, respectively. Excessive formaldehyde and other synthesis conditions were favored for the formation of polymers of different structures in the composites. An additional stage of high-temperature oxidative degradation was observed on the DSC curves, since the samples retained more water and contained more polymer.

The thermal analysis of the RFC-75 and FRC-78 composites (Figure 5) showed one main mass loss step (TG) with the corresponding thermal event in the DTG curves. This mass loss can be assigned to the carbon skeleton decomposition with gasification to $CO_2$, whereas the residue corresponds to dehydrated bentonite. The amount of carbon in the samples after carbonization varied in direct proportion to the polymer content in composites (Table S1). The amount of physically adsorbed water was less than 1 wt.% in the chars (the first step up to 300 °C), therefore, it could be ignored upon the calculations. TG and DTG curves of other samples were of the same type; therefore, they are not given here.

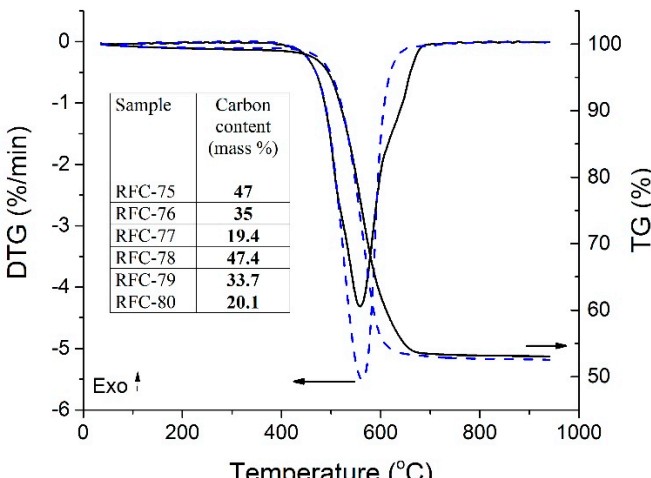

**Figure 5.** TG/DTG curves of the RFC-75 (straight line) and FRC-78 (dash line).

FTIR spectrometry combined with TG analysis provided information that allowed the qualitative identification of volatile products formed during thermal degradation. The FTIR spectra were recorded for evolved gases formed at 540–570 °C upon thermal decomposition of RFC samples (Figure 6). In this temperature range, the main gaseous product is $CO_2$ presented as a sharp peak at 2360 cm$^{-1}$ and a band of reduced intensity at 700 cm$^{-1}$. The bands at 2240–2060 cm$^{-1}$ are characteristic for CO produced due to incomplete combustion. It was mainly formed by the breaking of C–O–C and C=O [32]. However, all the carbon-containing compounds mentioned above can be further oxidized to

$CO_2$ with further temperature treatment. Weak bands corresponding to the OH stretching vibrations at 4000–3500 $cm^{-1}$ were related to water release.

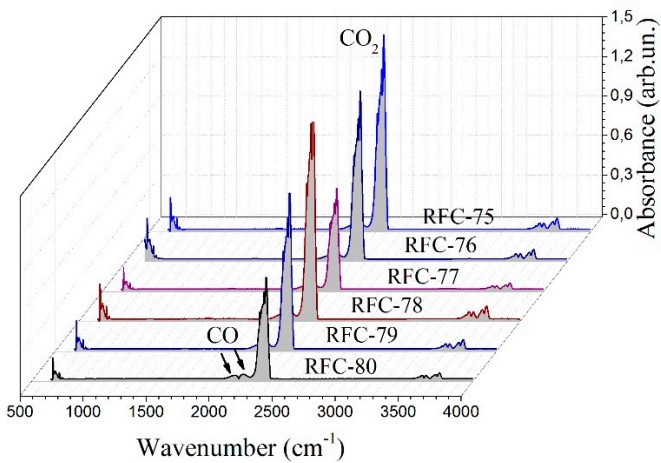

**Figure 6.** FTIR spectra of the carbonized composites heated at 540–570 °C.

## 4. Conclusions

Thus, traditional adsorbent bentonite has been modified using an efficient and inexpensive method to create new hybrid composites for further use as adsorbents (e.g., for various organics from wastewater). There is a strong influence of the synthesis method of initial composites on the morphological, textural, and structural characteristics of nanocomposites. The composites are characterized by different particle sizes, shapes, specific surface area and degree of graphitization after carbonization of RFR. Obtained results confirmed that particles of as-prepared composites have irregular shapes with developed micro- and meso-porosity. These characteristics show the effectiveness of the proposed approach to develop microporous and/or mesoporous structures in modified bentonite. From the results of DSC and TGA analysis, there was appropriate thermal stability of the carbon/bentonite composites. It was found that carbon/bentonite composites were characterized by larger specific surface area than that of initial bentonite due to developed mesoporous structures that are of importance from a practical point of view.

In addition, obtained organo-bentonite (particularly RFR 75–77) and carbon-inorganic composites can be considered as promising adsorbents. Untreated composites could acquire the properties of ion-exchange resin (due to a number of polar O-containing functionalities in both phases) with higher specific surface area values in comparison to bentonite alone. Carbon-inorganic composites are characterized by relatively high pore volume and specific surface area of micropores that favorably distinguishes them from bentonite alone.

**Supplementary Materials:** The following are available online http://www.mdpi.com/2504-5377/3/1/18/s1, **Table S1**: The ratio of the initial components used to prepare composites; **Table S2**: The specific surface area ($S_{SAXS}$) and the weight coefficients in the PaSD (model with lamellar (lam), cylindrical (cyl) and spherical (sph) particles) calculated using the self-consistent regularization procedure; **Figure S1**: Nitrogen adsorption–desorption isotherms (a, b, c) and (e) pore size distribution of initial bentonite; **Figure S2**: Normalized SAXS curves for initial RFR and related carbonized samples RFC; **Figure S3**: (a) Chord size distributions, and (b) particle size distributions (with a complex model of lamellar, spherical and cylindrical nanoparticles) calculated using the SAXS data with self-consistent regularization procedure, and: curves 1—RFR-75, 2—RFRC-75, 3—RFR-76, 4—RFC-76, 5—RFR-77, 6—RFC-76, 7—RFR-77, 8—RFC-78, 9—RFR-79, 10—RFC-79, 11—RFR-80, 12—RFC-80; **Figure S4**: SEM images of nanocomposites; **Figure S5**: Gram-Schmidt curves for the combustion of samples of first (a) and second (b) series.

**Author Contributions:** D.S. performed and assisted the analyses, helped with data interpretation, wrote the paper; M.G. and V.M.B. conceived and designed the experiments, analyzed and interpreted the data, wrote the paper; V.M.G. supervised the work, analyzed and interpreted the data and wrote the paper.

**Funding:** This research was funded by the Ministry of Education and Science of Ukraine, grant number: M/118–2018.

**Acknowledgments:** The authors are grateful to the Ministry of Education and Science of Ukraine for the financial support of the Project number: M/118–2018.

**Conflicts of Interest:** The authors declare no conflict of interest.

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
