# Peer review of "Influence of the Synthesis Method on the Structural Characteristics of Novel Hybrid Adsorbents Based on Bentonite"

_colloids, doi:10.3390/colloids3010018_

Round 1
Reviewer 1 Report
The manuscript (colloids-421901) reports the effect of synthesis method on the structural characteristics of bentonite/resorcinol formaldehyde resin composites as potential absorbents. The materials were then carbonized, and the influence of carbonization on the structural characteristics was investigated as well. Generally, this article is interesting and well written. I recommend it for publication in Colloids and Interfaces after the authors address the following concerns:
Line 70. Please add “The samples were labelled as RFC-75, RFC76, and RFC-77, respectively.”
Line 140. Please revise “Table 2” with “Table 1”.
The authors did not discuss the reason why different synthesis methods resulted in such distinctive structural characteristics of composites.
The purpose of carbonization is not clear.
After carbonization, why the specific surface area of the first series decreased; whereas the specific surface area of the second series increased?
What are the merits of this novel absorbent compared with traditional adsorbent?
Specific absorption experiments are not conducted.
In the introduction section, the authors can also consider some literature as references, e.g., Applied Clay Science, 2015, 118, pp 265-275, ACS Applied Nano Materials, 2018, 1 (12), pp 7039–7051; which reported the use of bentonite as reinforcing and viscosity modifiers in polymer composites and drilling fluids, respectively.
Author Response
We would like to thank the reviewer for the insightful comments on the manuscript, as these comments led us to an improvement of the work. Our revisions reflect all suggestions and comments. Detailed responses to reviewers are given below.
Point 1: Line 70. Please add “The samples were labelled as RFC-75, RFC76, and RFC-77, respectively.”
Response 1: This was added.
Point 2: Line 140. Please revise “Table 2” with “Table 1”.
Response 2: It was corrected.
Point 3: The authors did not discuss the reason why different synthesis methods resulted in such distinctive structural characteristics of composites.
Response 3: According to this point, as well as to Point 5, the additional explanations were added (Results and Discussions section, lines 151-162).
Point 4: The purpose of carbonization is not clear.
Response 4: The purpose of carbonization is to synthesize new hybrid samples while retaining the structural advantages of bentonite (in particular mesoporosity) and impart new structural qualities (microporosity) characteristic of carbon composites.
Additional description was added to Introduction (lines 61-65).
Point 5: After carbonization, why the specific surface area of the first series decreased; whereas the specific surface area of the second series increased?
Response 5: This comment along with Point 3 was described in the Results and Discussions section (lines 151-162).
Point 6: What are the merits of this novel absorbent compared with traditional adsorbent?
Response 6: Obtained organo-bentonite (particularly RRF 75-77) and carbon-inorganic composites can be considered as promising adsorbents. Untreated composites could acquire the properties of ion-exchange resin with higher specific surface values in comparison with bentonite. Carbon-inorganic composites are characterized by high pore volume and specific surface of micropores, which favorably distinguishes them from bentonite (Conclusions, line 276-281).
Point 7: Specific absorption experiments are not conducted.
Response 7: The structural characteristics of the adsorbents, as well the presence of a number of polar surface functionalities make it possible to approach the choice of adsorptives, as well as the area of their most effective application more reasonably. This will be carried out with the implementation of the development of this work.
Point 8: In the introduction section, the authors can also consider some literature as references, e.g., Applied Clay Science, 2015, 118, pp 265-275, ACS Applied Nano Materials, 2018, 1 (12), pp 7039–7051; which reported the use of bentonite as reinforcing and viscosity modifiers in polymer composites and drilling fluids, respectively.
Response 8: The recommended references were added.

Reviewer 2 Report
In the current manuscript entitled “Influence of synthesis method on the structural characteristics of novel hybrid adsorbents based on bentonite”, the authors have prepared hybrid composite materials by polymerization of resorcinol-formaldehyde resins in the presence of bentonite. While going through the manuscript, I have found the following deficiencies.
1. Authors should explain in details why they choose resorcinol-formaldehyde resins as organic compound of hybrid composite
(In Introduction)
2. Authors say in the lines 189-191:" For composites, the temperature range of polymer degradation becomes narrower in comparison to control polymer."
The following explaination not enough " This behavior might be explained by catalytic degradation of polymer in the presence of bentonite causing 100 °С decrease in decomposition temperature comparing to the initial polymer. For example, Florencio et al. reported [28] that the thermal degradation of polypropylene (in the solid state) filled by a clay was stronger than that of the pure polymer [28]." (line 190-194) (In Results and discussion.)
3. Authors should show more clear the obtained samples after thermal treatment carbonized (line 21) or graphitization (line 234).
Author Response
We would like to thank the reviewer for the insightful comments on the manuscript, as these comments led us to an improvement of the work. Our revisions reflect all suggestions and comments. Detailed responses to reviewers are given below.
Point 1: Authors should explain in details why they choose resorcinol-formaldehyde resins as organic compound of hybrid composite
Response 1: Additional information was added to Introduction (lines 51-60)
Point 2: Authors say in the lines 189-191:" For composites, the temperature range of polymer degradation becomes narrower in comparison to control polymer."
The following explaination not enough " This behavior might be explained by catalytic degradation of polymer in the presence of bentonite causing 100 °С decrease in decomposition temperature comparing to the initial polymer. For example, Florencio et al. reported [28] that the thermal degradation of polypropylene (in the solid state) filled by a clay was stronger than that of the pure polymer [28]." (line 190-194) (In Results and discussion.)
Response 2: Additional explanations were added (lines 219-229)
Point 3: Authors should show more clear the obtained samples after thermal treatment carbonized (line 21) or graphitization (line 234).
Response 3: Additional description and explanations were added. Note that used conditions (800 oC) do not allow the graphitization of the samples. The G band and G/D ratio analysis shows that the degree of the graphitization of the carbon phase is low.

Round 2
Reviewer 1 Report
The authors have addressed my concerns, and therefore I recommend it for publication in Colloids and Interfaces as is.
Reviewer 2 Report
In the current manuscript entitled “Influence of synthesis method on the structural characteristics of novel hybrid adsorbents based on bentonite” can be accept in present form.